# Peer review of "Optimizing Soluble Cues for Salivary Gland Tissue Mimetics Using a Design of Experiments (DoE) Approach"

_cells, 2022, doi:10.3390/cells11121962_

Round 1
Reviewer 1 Report
Overall, the paper describes a good experimental approach but I feel that most of their images are of low quality and they provide no immunostaining data to back up their claims that are solely based on RT-PCRs. Also, some figures were lacking control data and statistics.
P2L64: Is there a reason why DoEs haven’t been used? Are there disadvantages to this type of experiment which would preventing others from utilizing it? Is it time consuming or inefficient?
P2L90: You mention seven factors which you identified from the literature, they are listed elsewhere, but please list them here as well.
P13L370: You carried out these experiments for 14 days, and the preliminary experiments were 7 days in length. Would you be able to repeat these for longer periods of time to see how the factors are affected?
P19L519: it may be useful to perform this experiment using different types of salivary gland tissue to compare with the results of the study you mention?
Fig1D: It looks like image D has more background fluorescence than the other images. Please provide a better quality image .
Fig2,4,7,8: All these figures show the change in gene expression level via RTPCR, it might be worthwhile showing the actual protein expression levels as well to see if this translates at the protein level too.
Fig 6: What does the author mean by percentage mist1 expression? Is it the total percentage of cells expressing mist1 or is it a percentage relative to a housekeeping gene?
Fig 8: Please include the p values for the statistics in the figure legend.
Fig 9d-g: The figure shows the averaged response to the agonist, please include the error bars in the plot. Also are these plots representative plots of response to the agonists or are they averaged across multiple experiments? Please include the number of times these experiments were conducted and their statistics.
Fig 10: Please provide an image of the Day 0 control so as to provide visual comparison.
Fig 10i: Are these values statistically significant from each other? No statistics or number of repeats were mentioned.
Author Response
Thank you for your comments. Please see the responses and updates to the manuscript.
P2L64: Is there a reason why DoEs haven’t been used? Are there disadvantages to this type of experiment which would preventing others from utilizing it? Is it time consuming or inefficient?
As stated in lines 67-68. DoEs are more efficient than other methods. While we cannot guess why DoEs are not commonly used in academia, we hypothesize that it is due to a lack of knowledge and understanding of DoE methods. Statistically designed experiments offer clear advantages over traditional one-factor-at-a-time approaches and they are widely adopted in industry.
P2L90: You mention seven factors which you identified from the literature, they are listed elsewhere, but please list them here as well.
This has been updated – the current line number is 97.
P13L370: You carried out these experiments for 14 days, and the preliminary experiments were 7 days in length. Would you be able to repeat these for longer periods of time to see how the factors are affected?
The purpose of the preliminary experiments (ie. the Plackett-Burman and Box-Behnken DoEs) was to identify factors that promote Mist1 expression. It is unlikely that factors would promote Mist1 expression at day 14, but not at day 7. Thus we don’t foresee the results changing with the additional experiments suggested. We believe the decline in Mist1 expression at day 14 underscores the need for complete optimization of culture conditions – we are currently working on optimizing the hydrogel matrix and combining this with the work presented here to address this, as mentioned in lines 661-662.
P19L519: it may be useful to perform this experiment using different types of salivary gland tissue to compare with the results of the study you mention?
Testing on a different salivary gland tissue model is out of the scope of this paper. We hope that other groups adopt our results as it would be interesting.
Fig1D: It looks like image D has more background fluorescence than the other images. Please provide a better quality image .
This is not background fluorescence – the ROCK inhibitor induces cell proliferation, so there are cells on the surface of the chip that are being stained with calcein AM. This is further explained in Figure S1A.
Fig2,4,7,8: All these figures show the change in gene expression level via RTPCR, it might be worthwhile showing the actual protein expression levels as well to see if this translates at the protein level too.
We agree this would be worthwhile and is an area of current investigation.
Fig 6: What does the author mean by percentage mist1 expression? Is it the total percentage of cells expressing mist1 or is it a percentage relative to a housekeeping gene?
Gene expression is relative to the housekeeping gene Rps29 and normalized to day 0 – the percentage is compared to day 0. This is described in the Methods section, 2.10 RNA extraction and qPCR.
Fig 8: Please include the p values for the statistics in the figure legend.
This has been updated.
Fig 9d-g: The figure shows the averaged response to the agonist, please include the error bars in the plot. Also are these plots representative plots of response to the agonists or are they averaged across multiple experiments? Please include the number of times these experiments were conducted and their statistics.
In section 2.11, the methods have been updated to clarify that the traces are the averaged response to the agonist for n = 3 experiments. Given that images are taken every 1 second, error bars are not included as it would make it impossible to distinguish different points on the graph. Similarly, it would not make sense to show statistics on every point. The way we plotted it is similar to others in literature, cited in lines 254-255. To view statistical comparison on calcium signaling data, please see duration and latency results in Figures S8 and S9.
Fig 10: Please provide an image of the Day 0 control so as to provide visual comparison.
The Day 0 control image is shown in Figure S10.
Fig 10i: Are these values statistically significant from each other? No statistics or number of repeats were mentioned.
This figure has been updated with statistics.
Reviewer 2 Report
The manuscript titled “Optimizing soluble cues for salivary gland tissue mimetics using a design of experiments (DoE) approach” describes the design of experiments approach to improve primary salivary gland cell culture in vitro. Overall, the manuscript is well-written and prepared for the most part. However, the followings are some concerns and comments have been pointed out that the authors may want to consider.
1. Line 27 Keywords: The keyword “media optimization” does not appear in the main context. The “salivary glands” does not appear in the main context, but “salivary gland”. The “tissue engineering” only appears one time in the main context, in line 47. The “soluble factors” only appears one time in the main context, line 514. I do not think they are suitable to be the keywords. Please consider switching them to more suitable keywords.
2. Line 50, line 112, and line 128: Remove unnecessary self-citations. Keeping one of your previous publications should be enough.
3. Lines 73-75: I’d suggest inserting references here “Mist1 expression was…..of FGF10 and neurturin.”
4. Line 84 Materials and Methods section: Please provide reagents that were used in this study with detailed information to make your work reproducible.
5. Line 91: Please provide the reference to “…identified from literature”.
6. Line 118, “weight%”; line 126, “wt%”. Please homogenous the format, if the authors prefer “wt”, please define it before using it.
7. Line 129: The incubation condition should be mentioned. For example, temperature, etc.
8. Line 189: The authors stated, “Cells were cultured for 7 days” (line 152, line 166, and line 181). Why did the authors list 14 days of the culture here? Please specify 14 days of cell culture. Additionally, do the authors have any idea about how long primary salivary gland cells can be maintained in good condition in vitro? If yes, I’d suggest the authors provide this information in the context that might be very helpful to other researchers.
9. Line 199: Table S1 and Table S2 are overlapped. Please separate them to make the tables are readable.
10. Line 232: Please specify “ANOVA”, one-way? Two-way?
11. Line 233: Please rephrase “t-tests for individual factor p-values” to make it easier to understand.
12. Line 233: I’d suggest the authors use italic p as it refers to a p-value. Check throughout the manuscript.
13. Line 250 Figure 1: a) Please label the treatment with each image for easier tracking and reading. b) I’d suggest the authors add scale bars/lines with double arrow end to indicate how the spheres’ sizes have been measured. That might be very helpful to other researchers to follow your protocol.
14. Line 261 and Figure 3, Figure 5: Please provide details that how to predict values for the model.
15. Line 268 Figure 2 and line 304 Figure 4: a) The run number for X-axis is missing. b) I’d suggest the authors use “Relative … mRNA expression…” as the title for Y-axis.
16. Line 281: I’d suggest the authors add “p values” to Figure 3A-C in each image instead of only in the context.
17. Line 281 Figure 3, line 325 Figure 5: Please provide at least a brief statistic description in the figure legend.
18. Line 339 Figure 6, line 358 Figure 7, and line 393 Figure 8: I’d suggest the authors use “Relative … mRNA expression…” as the title for Y-axis.
19. Line 358 Figure 7: The statistic description is not clear, please rephrase.
20. Line 393 Figure 8: The statistic description in the figure legend is missing. Please provide.
21. Line 417 Figure 9: The scale bar in Figure 9B is missing.
Author Response
Thank you for your thorough reviewer and useful comments. Please see our point-by-point responses.
- Line 27 Keywords: The keyword “media optimization” does not appear in the main context. The “salivary glands” does not appear in the main context, but “salivary gland”. The “tissue engineering” only appears one time in the main context, in line 47. The “soluble factors” only appears one time in the main context, line 514. I do not think they are suitable to be the keywords. Please consider switching them to more suitable keywords.
The keywords have been changed to: salivary gland; design of experiments; Mist1; acinar cell; EGFR inhibitor
- Line 50, line 112, and line 128: Remove unnecessary self-citations. Keeping one of your previous publications should be enough.
These have been updated to include one citation from our lab for each statement.
- Lines 73-75: I’d suggest inserting references here “Mist1 expression was…..of FGF10 and neurturin.”
This statement is a result of the current work and thus there are no references to cite.
- Line 84 Materials and Methods section: Please provide reagents that were used in this study with detailed information to make your work reproducible.
Vendors and catalog numbers have been added for the reagents. Lines: 109-110, 118, 132-133, 150-154, 215-216, 218-219, 221-222, 256-257
- Line 91: Please provide the reference to “…identified from literature”.
References were added – current line number is 97. These are also in Table 1.
- Line 118, “weight%”; line 126, “wt%”. Please homogenous the format, if the authors prefer “wt”, please define it before using it.
This has been updated to “weight%” for both. New line #131
- Line 129: The incubation condition should be mentioned. For example, temperature, etc.
The temperature has been added. New line #135
- Line 189: The authors stated, “Cells were cultured for 7 days” (line 152, line 166, and line 181). Why did the authors list 14 days of the culture here? Please specify 14 days of cell culture. Additionally, do the authors have any idea about how long primary salivary gland cells can be maintained in good condition in vitro? If yes, I’d suggest the authors provide this information in the context that might be very helpful to other researchers.
The experimental section details how many days the cells were cultured for each experiment.
For cell viability, the Plackett-Burman DoE, and the Box-Behnken DoE (sections 2.7, 2.8, and 2.9), cells were cultured for 7 days (Figures 1-6). For Figure 7-8, cells were cultured for 2, 7, and 14 days for analysis by qPCR. For Figures 9-10, cells were cultured for 7 and 14 days.
The following statement was added to line 55-56: “Our previous study showed that cells could be cultured for up to 14 days in vitro, but expression of key markers such as Mist1 dropped significantly.”
- Line 199: Table S1 and Table S2 are overlapped. Please separate them to make the tables are readable.
These tables are in the supplemental material document. This has been updated to a PDF file to help prevent formatting issues.
- Line 232: Please specify “ANOVA”, one-way? Two-way?
This has been updated to one-way ANOVA, new line #269
- Line 233: Please rephrase “t-tests for individual factor p-values” to make it easier to understand.
This has been updated to: “and t-tests to determine the effects of individual factors” Line 266
- Line 233: I’d suggest the authors use italic p as it refers to a p-value. Check throughout the manuscript.
This has been updated.
- Line 250 Figure 1: a) Please label the treatment with each image for easier tracking and reading. b) I’d suggest the authors add scale bars/lines with double arrow end to indicate how the spheres’ sizes have been measured. That might be very helpful to other researchers to follow your protocol.
Figure 1 was updated with the labels. Figure S1C was added to show how the sphere size was calculated and additional text was added to the Materials and Methods section under 2.7.
- Line 261 and Figure 3, Figure 5: Please provide details that how to predict values for the model.
The values were predicted based on the equation generated by JMP to fit the data, shown in Table S3 (Plackett-Burman) and Table S6 (Box-Behnken). Lines 299 and 359-360 have been updated to explain this.
- Line 268 Figure 2 and line 304 Figure 4: a) The run number for X-axis is missing. b) I’d suggest the authors use “Relative … mRNA expression…” as the title for Y-axis.
Both comments have been updated.
- Line 281: I’d suggest the authors add “p values” to Figure 3A-C in each image instead of only in the context.
The p-values are listed in the figure caption.
- Line 281 Figure 3, line 325 Figure 5: Please provide at least a brief statistic description in the figure legend.
Lines 329-331 and 380-382 have been updated to include statistic information.
- Line 339 Figure 6, line 358 Figure 7, and line 393 Figure 8: I’d suggest the authors use “Relative … mRNA expression…” as the title for Y-axis.
Figures 6, 7, and 8 were updated with the proper y-axis.
- Line 358 Figure 7: The statistic description is not clear, please rephrase.
Line 419 has been updated to “Statistics were calculated using two-way ANOVA”
- Line 393 Figure 8: The statistic description in the figure legend is missing. Please provide.
Lines 456-457 have been updated as follows: “Statistics were calculated using two-way ANOVA. * p < 0.05, ** p < 0.01, *** p < 0.001, **** p < 0.0001.”
- Line 417 Figure 9: The scale bar in Figure 9B is missing.
This has been updated.
Reviewer 3 Report
This study analyses a combination of seven soluble cues that were previously shown to maintain or improve salivary gland cell function. This was done to assess the utility and appropriateness of functional in vitro models by using a design of experiments (DoE) approach to test combinations of seven soluble cues.
This is a highly interesting study that is well written and has many scientific merits.
A figure summarizing the main effects per days and modifications to the culture medium and the main take-away points would greatly aid in improving clarity of concepts and help disseminate key information for other researchers.
Author Response
Thank you for your comments.
Response to Reviewer 2 Comments:
This study analyses a combination of seven soluble cues that were previously shown to maintain or improve salivary gland cell function. This was done to assess the utility and appropriateness of functional in vitro models by using a design of experiments (DoE) approach to test combinations of seven soluble cues.
This is a highly interesting study that is well written and has many scientific merits.
A figure summarizing the main effects per days and modifications to the culture medium and the main take-away points would greatly aid in improving clarity of concepts and help disseminate key information for other researchers.
A graphical abstract has been created to summarize the take-home messages.
Round 2
Reviewer 2 Report
I do not have any further concerns now, except for the following minor comments that the authors should consider and double-check to homogenous the format throughout the manuscript again before publication. Good luck.
1: Figure 1 legend and Figure 2 legend: It seems the fonts are not the same. Please double-check.
2: Lines 282-287 Figure 2 legend and Figure 2: Please make sure the figure and its legend are combined together.
3: Figure 1 legend: "," followed (A) to (F), "." followed (G) and (H).
Author Response
Thank you for your comments. Please see the our responses.
1: Figure 1 legend and Figure 2 legend: It seems the fonts are not the same. Please double-check.
The fonts are the same.
2: Lines 282-287 Figure 2 legend and Figure 2: Please make sure the figure and its legend are combined together.
This has been updated.
3: Figure 1 legend: "," followed (A) to (F), "." followed (G) and (H).
There are commas after A-F to indicate a list. There are periods after G and H to end the sentences.